# New Insights of Oral Colonic Drug Delivery Systems for Inflammatory Bowel Disease Therapy

**DOI:** 10.3390/ijms21186502

**Published:** 2020-09-05

**Authors:** Adrian H. Teruel, Isabel Gonzalez-Alvarez, Marival Bermejo, Virginia Merino, Maria Dolores Marcos, Felix Sancenon, Marta Gonzalez-Alvarez, Ramon Martinez-Mañez

**Affiliations:** 1Instituto Interuniversitario de Investigación de Reconocimiento Molecular y Desarrollo Tecnológico (IDM), Universitat Politècnica de València, Universitat de València, 46022 Valencia, Spain; adherte@upv.es (A.H.T.); Virginia.Merino@uv.es (V.M.); mmarcos@qim.upv.es (M.D.M.); fsanceno@upvnet.upv.es (F.S.); rmaez@qim.upv.es (R.M.-M.); 2CIBER de Bioingeniería, Biomateriales y Nanomedicina (CIBER-BBN), 46022 Valencia, Spain; 3Engineering Department, Pharmacy and Pharmaceutical Technology Section, Miguel Hernandez University, 03550 Alicante, Spain; isabel.gonzalez@goumh.umh.es (I.G.-A.); mbermejo@umh.es (M.B.); 4Pharmacy and Pharmaceutical Technology and Parasitology, University of Valencia, 46100 Valencia, Spain; 5Departamento de Química, Universitat Politècnica de València, Camí de Vera s/n, 46022 Valencia, Spain

**Keywords:** intestinal permeability, colon, drug delivery, inflammatory bowel diseases

## Abstract

Colonic Drug Delivery Systems (CDDS) are especially advantageous for local treatment of inflammatory bowel diseases (IBD). Site-targeted drug release allows to obtain a high drug concentration in injured tissues and less systemic adverse effects, as consequence of less/null drug absorption in small intestine. This review focused on the reported contributions in the last four years to improve the effectiveness of treatments of inflammatory bowel diseases. The work concludes that there has been an increase in the development of CDDS in which pH, specific enzymes, reactive oxygen species (ROS), or a combination of all of these triggers the release. These delivery systems demonstrated a therapeutic improvement with fewer adverse effects. Future perspectives to the treatment of this disease include the elucidation of molecular basis of IBD diseases in order to design more specific treatments, and the performance of more in vivo assays to validate the specificity and stability of the obtained systems.

## 1. Introduction

The development of new colonic drug delivery systems (CDDS) has gained interest in the last decades, since it was recognized that the colon shows several advantages as a local target, to improve the outcomes for different drugs and pathologies. CDDS are especially advantageous to achieve stable plasma levels from controlled release formulations to deliver protein- and peptides-based drugs (e.g., insulin, calcitonin, and vasopressin). These should be protected from proteolytic enzyme activity that is present along the gastrointestinal tract (GIT) and in the colon, where it finds a better environment for enhanced absorption and for targeting the colon against disorders or diseases that requires local treatment for reducing systemic adverse effects (e.g., irritable bowel syndrome, inflammatory bowel diseases, colorectal cancer, etc.) [1,2]. The main therapeutic advantages that CDDS can provide for local treatment of colonic diseases are (i) high drug concentration in the colonic inflamed tissues that implies better therapeutic effects, (ii) a reduction of adverse effects by a decreased systemic absorption of active agents, (iii) smaller amounts of active agents necessary to reach desirable therapeutic effects since high local concentrations is attained, (iv) a major compliance of patients due to the comfort of the oral route and the reduced frequency of drug administration, and (v) saving health economic resources [3,4]. In order to achieve specific site targeting, many strategies were developed by taking into account the specific features of the gastrointestinal tract. Moreover, permeability studies carried by Nakai et al. [5], which are standard permeability drugs in ulcerative colitis tissues and in normal intestinal tissues, revealed that drug permeability is reduced in inflamed tissues as a result of morphological changes in tissues. This fact confirms that colon release systems are particularly suitable for the treatment of these diseases.

Current clinical treatment of inflammatory bowel diseases (described in Section 3) depends on the severity of symptoms. The oral or parenteral routes are the most common and, through them, the drugs reach systemic levels. When looking for local effects, suppositories or rectal foams with the same active substances are used. However, there is no definitive treatment for these pathologies and patients experience outbreaks throughout their lifetime, which significantly limits their quality of life. Sometimes the severity of symptoms is disabling for the patient. In addition, a non-negligible percentage of patients present an outbreak of hospitalization required and they are treated with an aggressive combination of drugs or even surgery. The treatment of inflammatory pathologies can and should improve, and many research groups work on the development of new therapies. One of the most powerful lines of work is the development of colon-targeted release systems to obtain higher drug levels in inflamed tissue.

The focus of this review was to summarize the contributions reported in the last four years of the use of oral CDDS, to improve the effectiveness of treatments of IBD. The review focuses mainly on the development of CDDS, in which drug release was induced by pH, the presence of specific enzymes or the reducing environment of the colon. Some examples also show a combination of these approaches to develop CDDS for IBD.

## 2. Inflammatory Bowel Disease: Ulcerative Colitis and Crohn’s Disease

Inflammatory bowel diseases (IBD) are autoimmune, chronic relapsing inflammatory diseases that affect the gastrointestinal tract (GIT) and cause epithelial injuries. The intestine contains a huge antigenic burden derived from the diet and the intestinal flora. Most bacteria that are part of the intestinal flora are beneficial to health, as they collaborate in the digestive process and prevent colonization by pathogenic species. It seems that IBD appears when an abnormal response to usual flora occurs. Due to an overreaction of the immunity system, GIT cells are attacked by the immunity system, resulting in chronic inflammation and several damages. The translocation of luminal antigens (for example, bacterial antigens from the commensal microbiota) into the bowel wall, produced by alterations in epithelial barrier, increases the damage and induces inflammatory response. The term inflammatory bowel disease (IBD) is used to include the set of diseases that develop inflammation in some section of the intestine. Ulcerative colitis (UC) or Crohn’s disease (CD) are the most common types of IBD. Improved diagnostic, imaging and genotyping techniques enabled a better understanding of these diseases. However, the etiology of the disease is unclear. The development of these diseases is associated with a combination causes of immune dysfunction, genetic predisposition, and environmental factors [6,7,8]. The occurrence of only one of these causes is not enough for the development of the disease. Experimental and genome-wide association study results showed different mechanisms of immune system impairment, such as pro-inflammatory pathways led by IL-23, decreased mechanisms regulating immunity, and a defect in the barrier function of the intestinal epithelium. Pro-inflammatory cytokine TNF plays a crucial role in the process of intestinal inflammation, so the use of monoclonal antibodies is one of the therapeutic strategies, but is only used as a last option. Figure 1 shows some key cellular populations and mediators that could play a role in in intestinal inflammation.

UC and CD has similar symptoms, such as abdominal cramps and pain, bloody diarrhea, weight loss, anemia, general unrest, and some similar physiopathological characteristics, such as intestinal tissue inflammation, ulcers, or even fissures. These coincidences make it difficult to distinguish between them without a specific diagnostic study.

The main difference between them is the GIT segment that is affected by the lesions. In the case of ulcerative colitis, it only affects the colorectal mucosa, inflammation is superficial, and does not affect other intestinal layers. In Crohn’s disease, inflammation affects any section of the digestive tract, although the ileum and colon are often the most affected segments. Inflammation appears discontinuously so that the healthy tissue areas alternate with the inflamed tissue areas. Moreover, in CD, inflammation might affect all layers of the intestine (transmural). Sometimes in CD, symptoms like sores in the mouth, anal fissures, or narrowing that are infrequent in UC patients appears (Figure 2).

Currently, the differentiation between the two pathologies has improved with the new methods of diagnosis, such as endoscope diagnosis or imaging techniques. Moreover, serological markers are also useful to differentiate between the two pathologies. For this purpose, levels of anti-saccharomyces cerevisiae (ASCA) antibodies and perinuclear anti-neutrophil cytoplasmics (pANCA) are determined. pANCA are mainly found in patients with ulcerative colitis (60–80%), while ASCA is found mostly in patients with Crohn’s disease (45–60%).

Both diseases, UC and CD are chronic, currently not curable, and are subject to remissions and relapses. This has a clear impact on patients’ quality of life, therefore, making it necessary for us to find a more efficient therapy. Colon-specific delivery systems are the best options for treating these local disorders.

## 3. IBD Treatments

Most treatments available for IBD focus on remission of symptoms, and the selection of treatment depends on the phenotype and severity of the exacerbation disease. When the disease develops mildly or moderately, the drug of choice is 5-aminosalicilic acid (5-ASA), also known as mesalazine or mesalamine. However, oral administration of 5-ASA in a fast-release dosage form is generally not always the best option, because it is absorbed into the proximal small intestine and metabolized without reaching the therapeutic levels in inflamed tissues. 5-ASA is only useful as a local therapeutic agent and the plasma levels have no therapeutic effects. For this reason, other aminosalicylates containing 5-ASA, such as sulfasalazine, olsalazine, and balsalazide (Figure 3) are normally administered orally and release 5-ASA under specific conditions, in order to optimize the administered dose and treat IBD more efficiently. Sulfasalazine was the first 5-ASA designed prodrug that bound 5-ASA to sulfapiridine through an azo link. About 90% of oral sulfasalazine reaches the colon intact and, once there, the azoreductase enzymes produced by anaerobic microbiota break the azo bond and release the 5-ASA in colon. In this way, the drug reaches therapeutic levels in the colon. Sulfapyridine is additionally absorbed, acetylated, hydroxylated, and conjugated with glucuronic acid in the liver, and is finally removed by the kidneys. Sulfapyridine does not produce therapeutic effects, but instead it is responsible for most of the adverse effects of sulfasalazine.

The choice of treatment to induce remission for more severe disease symptoms (or when aminosalicylates are contraindicated) are corticosteroids like hydrocortisone, budesonide, prednisolone, or beclomethasone. Rectal administration of corticoids provides a topical treatment of IBD diseases, which is especially useful for ulcerative colitis as UC rectum is always affected by inflammation spreading from the distal to the proximal colonic segments. Corticosteroids are formulated as suppository, rectal cream, foam, or liquid formulations, in order to allow rectal administration. The main advantages of this route is the increasing drug level in the inflamed tissues and the reduction of plasmatic levels and adverse effects associated with corticosteroids. However, patients are reluctant to use the rectal path and, moreover, rectal formulations have some therapeutic limitations to treat IBD, as the drug distribution in inflamed areas is not homogenous [10,11] and only reaches the rectal area and parts of the colon [3,12]. The most common alternative is the oral route corticosteroid administration. However, this route of administration is most likely to cause significant side effects as high plasma levels are reached. Depending on the dose, immunosuppressive effects (increased risk of infections), glaucoma or cataracts, high blood pressure, osteoporosis, metabolic dysfunctions of bruising, and slower wound healing in the skin, etc. could be observed. Specific targeted formulations (described in this review and other) are required to overcome these limitations. For severe disease relapses, a combined therapy of intravenous corticosteroids or immunosuppressive drugs like azathioprine, 6-mercaptopurine, methotrexate, tacrolimus, cyclosporine, and calcineurin inhibitors is necessary.

These drugs groups (aminosalicilates, corticosteroids, and immunosuppressive agents) used for the treatment of IBDs are collectively referred to as pre-biological therapeutic options. Nowadays, there are new therapeutic options called biological therapies, based on biological molecules like anti-TNF agents (infliximab, adalimumab, golimumab, and certolizumab) or anti-integrin agents (natalizumab and vedolizumab) [13]. The main advantage of biological agents is their ability to maintain remission and to achieve mucosal healing without using corticosteroids. However, all these biological drugs, as well as other immunosuppressive drugs, also have adverse effects, as they reduce the ability to fight infections and therefore increase the probability of suffering severe viral, bacterial, or fungal infections that could spread throughout the body. CDDS are the best therapeutic option to take advantage of the dose, increase the therapeutic benefits, and reduce the adverse effects.

Finally, if pharmacological therapies are not enough or fulminate disease stages appear, IBD patients might require surgery [4,14]. In CD cases, the disease might reemerge but not in UC patients. Nonetheless, this surgery (colectomy) brings a decrease in the wellbeing of patients.

## 4. IBD Physiology and Microbiota Population

Colonic-targeted delivery systems are designed to release drugs specifically in the colon and not before. For this reason, it is of great importance to take into account the peculiarities of the digestive tract with regards to transit time, pH, microbiota, osmotic pressure, etc., in healthy and disease conditions, while designing these drugs.

Gastrointestinal transit varies, depending on several factors like stomach transit time, which takes between <1 h when fasting or >3 h in non-fasting conditions. Small intestinal transit time generally varies between 2 to 6 h in healthy subjects but this time is delayed (≈30%) in IBD patients [15,16,17,18]. Moreover, colonic transit times ranges from 6 to 70 h reported in healthy patients, whereas in IBD patients this is significantly faster, likely because of diarrhea—a hallmark of IBD diseases [15,19,20,21,22].

pH values in healthy individuals start range from pH ≈ 1–5 (acid pH in the stomach) increasing along the small intestine till pH ≈ 7.4, at the terminal ileum. Then, pH decreases in the cecum, pH ≈ 5.5–6, and finally it rises again, pH ≈ 6.7 at the rectum. Nevertheless, pH ranges can vary, intra or interindividually, with water or food intake, and microbial metabolism, but especially in the presence of active IBD [11]. In particular, it was reported that important pH decreases affect the colon in both UC and CD (pH ≈ 2.3–5.5) [15,23,24,25,26,27].

Microbiota of the stomach and the small intestine mainly consists of gram-positive facultative bacteria (103–104 CFU/mL) [28,29,30]. Moreover, local colon microbiota is mainly composed of anaerobic bacteria (e.g., Bacteroides, Bifidobacteria, Eubacteria, Clostridia, Enterococci, Enterobacteria, etc.) and is many times greater (1011–1012 CFU/mL) [31]. The main metabolic function of the flora is the fermentation of the residues of the non-digestible diet and mucus, produced by the intestinal epithelium [32]. To carry out this process, microbiota produces a great quantity of enzymes such as ß-galactosidase, azoreductase, ß-xylosidase, nitroreductase, glycosidase deaminase, etc. [33,34]. Some changes in microbiota composition (intestinal dysbiosis) are common in GI diseases like IBD, with alterations in pH, inflammatory process, bacterial enzymatic metabolism and physiology [15,35,36].

These specific features of the gastrointestinal transit and the microbiota are used for the design of CDDS (vide infra) in which a specific change in the pH or the activity of certain types of enzymes are used as triggers for the release of the drug.

Another important aspect to reach the colon epithelium is the proper particle size of drugs or their carriers. Particles with diameter higher than 200 µm have low GIT transit times due to the physiological conditions of the IBD diseases in the injured intestine. However, it was reported that micro and nanoparticles could introduce some advantages in targeting the inflamed areas in colon, due to their smaller particle size, as they could pass through the GIT easier than classical single unit dosage forms [34,37]. Nanoparticles (NPs) have the characteristic of enhancing epithelial permeability and retention in inflamed areas (similar to the enhanced permeability and retention (EPR) effect observed in tumor tissues). For this reason, immune cells, highly expressed at inflamed regions, develop a preferential uptake of nanoparticles [15,27,38,39]. In addition, particles of nanoscopic diameter are not swept away by feces, even when diarrhea occurs in IBD [40]. This ability of nanoparticles to reach the colon and remain in this target organ while releasing the drug into inflamed tissue determined the research focus on the development of increasingly sophisticated and effective nanoscopic systems. However, the systemic absorption that occurs when using these systems should also be studied, given their high time of residence in the colon. In fact, some studies showed that the absorption of drug could become important and that, in addition, the nanoparticles themselves can be absorbed in the colon and rectum of patients with IBD [41]. They reported the accumulation of microparticles (MPs) in active inflammation, but negligible amount of nanoparticles could be detected [23,42]. This meant that MPs, probably due to their bioadhesion, could be accumulated in the inflamed mucosal wall, but they could not be absorbed. However, when particles reach systemic circulation, they are recognized by the mononuclear phagocyte system and they are taken to the liver and spleen, limiting the efficacy of the safety profile of the nanoformulation. This study suggested that NPs should not be required for local drug delivery to intestinal lesions in humans [42]. On the contrary, in vivo studies showed an increased smaller particle size deposition in animals with induced ulcerative colitis relative to healthy ones. The reason for the discrepancy is not yet elucidated. Some authors suggested that transport across inflamed intestinal mucosa is different, depending on size particles and indicate that nanoparticles can reach the deeper layers of the mucosa, while micro-sized particles stay in the surface layers [42,43]. Therefore, MPs could be administered in active disease, whereas NPs could be administered in minor inflammation cases when the integrity of the mucosal barrier is less affected, and absorption through epithelial cells is the most important mechanism.

A definite continuous effort is being made to get targeted drug delivery systems to prevent or treat IBD, even modifying the size of the delivery systems, according to the severity of the disease. The final aim is always to increase the desired therapeutic effects of the existing or novel drugs, decreasing the adverse effects as much as possible. The longer lasting permanence of the drug locally within the affected epithelium in colon would allow the reduction in drug amounts and frequency of intakes, thereby increasing the treatment adherence. Moreover, safety and efficacy could be improved by administration of biological drugs focused on inflamed tissues.

## 5. Oral CDDS for IBD Management

For several decades, different strategies were used to target the colon, such as biodegradable polymers that respond to changes in pH, time-dependent formulations, pressure-controlled drug delivery, polysaccharide-based formulations, osmotic-controlled systems, bioadhesive systems, inflammation targeting, microbiota, and enzyme responsive prodrugs, or drugs coated with sensitive polymers. These strategies are already described in numerous articles and reviews [44,45,46,47].

### 5.1. Oral CDDS Following the pH-Sensitive Approach

It is well-known that the pH of the gastrointestinal tract varies from 1.2 (stomach) to 7.5–8 (large intestine). Therefore, pH-sensitive polymers were the first strategy to achieve colonic drug delivery. Enteric polymers that are most commonly used are copolymers of metacrylic acid and methyl metacrilate (Eudragit^®^ S, Eudragit^®^ L, Eudragit^®^ FS, Eudragit^®^ L30D-55, Eudragit^®^ P4135F) HPMCAS-LF (hypromellose acetate succinate), HP50 (hydropromellose phthalate50), and HP55 (hydropromellose phthalate55) [48,49,50]. These materials could be used to coat tablets, capsules, matrix, liposomes, and so on, in order to successfully encapsulate drugs with different lipophilia [49,51].

As an example of this approach, Naeem et al. [42] encapsulated cyclosporine A in Eudragit FS30D nanoparticles (ENPs), poly(lactic-co-glycolic acid) (PLGA) nanoparticles (PNPs), and Eudragit FS30D/PLGA nanoparticles (E/PNPs) obtained through the oil-in-water emulsion method. The combination of these coatings allowed to reduce drug release (only 18%) at acidic pH, until values 6–8 were reached, and increased the release at colonic pH. In vivo assays in a mouse model of the disease, revealed an improvement of the main markers, such as colon length, weight loss, rectal bleeding, spleen weight, histological scoring, and inflammatory indicators (myeloperoxidase activity, macrophage infiltration, and expression of proinflammatory cytokines), after E/PNP nanoparticle administration, compared to nanoparticles with only one coating [42]. Natural-derived polymers such as alginates are also widely used in this field. For instance, Oshi et al. [43] developed a new formulation based on dexamethasone microcrystals (DXMCs), coated with several layers, using a layer-by-layer coating technique. Coatings of chitosan oligosaccharide (CH), alginate (AG), and finally Eudragit S 100 (ES) were added in order to achieve colon-targeted delivery. This approach allowed us to obtain a pH-dependent dexamethasone release, reducing the initial burst of drug release at pH 1–2 and 6–8 that corresponds to the stomach and small intestine conditions, and providing sustained dexamethasone release at pH 7–4, which corresponds to the colonic conditions. In vivo assays carried out in a mouse model of colitis showed that the oral administration of this formulation resulted in a better therapeutic activity, compared to other dexamethasone formulations. Bazan et al. [52] developed celecoxib-loaded Eudragit microparticles and they carried out in vivo assays in a rat model. Results indicated that the microparticles formulated using Eudragit S100 and, to a lesser extent Eudragit L100-55, provided a controlled release of celocoxib, important reduction of colonic injury, and a decrease of colon inflammation and inflammation markers.

Most products approved for the specific treatment of IBDs, such as Asacol^®^ (Eudragit S^®^), Claversal^®^ (Eudragit L^®^), Mezavant/Lialda^®^ (multi-matrix tablet coated with (Eudragit S^®^), Octasa^®^, Apriso^®^ (Eudragit L^®^ coated granules), or Salofalk^®^ (Eudragit L^®^) are based on this strategy. These formulations are the first-line treatment for the treatment of moderately intense inflammatory colon diseases [53,54,55]. A limitation of the pH-sensitive approach is, as commented before, the changes in the pH of the GIT in a non-defined and unexpected way in IBD patients, as well as the inter- and intravariability in patients and the fast or fed condition of subjects that can also have an influence on pH [15,22,23,27]. In vivo studies carried out in human volunteers revealed that similar 5ASA metabolite concentrations were detected in feces, indicating that similar doses were available for treatment [56]. Moreover, in vivo studies indicate that Eudragit S-coated tablets do not exhibit site-selectivity drug delivery due to many physiological factors like feed status, colon pH, or transit time [57].

Mucoadhesive systems allow to increase the residence time in colon and it is useful to overcome limitations related to accelerated transit time [55,58,59,60]. Agüero et al. [59] carried out an important review highlighting the properties of alginate microparticles as a colon drug delivery system, using the oral route of administration. Alginate (a linear polymer of mannuronic and glucuronic acids, found in the cell walls of algae) presents pH-sensitiveness, mucoadhesiveness, biocompatibility, and gelling ability, which allows us to design effective, targeted colon systems, in order to improve the therapeutic options of IBD patients. Polymers such as alginate or chitosan, exhibit mucoadhesive properties. Cong et al. [61] obtained a pH-sensitive drug delivery system based on a matrix composed of alginate hydrogel and chitosan micelles. The results reflected that there are several mechanisms implicated, consecutively or simultaneously, in the release profile of simulated GIT fluids—sustained-release profile for hydrogel/micelle (1:1) and colon-specific profile for hydrogel/micelle (3:1). pH-sensitive systems are very useful and work well, but generally the release occurs quickly when the pH change occurs. Other new technologies were developed in recent years, such as Colopulse, which consist of a coating matrix that includes a super-disintegrant for accelerating disintegration in the target. In vivo studies demonstrated the site-specificity in healthy subjects and IBD patients, independent of the food intake. This technology allows us to overcome limitations due to an accelerated transit time in IBD patients and is used to developed budesonide tablets and infliximab tablets [62,63]. Other examples following this approach are summarized in Table 1.

Despite pH-sensitive approach systems offering important advantages for IBD treatments, new combined strategies were developed to achieve more accurate, site-specific delivery, independent of pH variability, transit time, or feeding condition.

### 5.2. Oral CDDS Following Enzyme-Sensitive Approach

In recent years, many enzyme-sensitive systems were developed, in order to take advantage of the specific colon microbiota for targeted release in colon. For this purpose, polymeric nano and microsystems were developed using degradable polymers through specific colon enzymes, such that, when the polymer was degraded, the drug was released locally. Some prodrugs were obtained using the same approach, so they become the active drug, after a bond rupture carried out by the action of specific colonic enzymes. In both cases, the excision of specific bonds, such as the azo group, involves the targeted release of the drug. A typical example is the non-hydrolyzed starch that ferments upon reaching the colon, thanks to the colon anaerobic microbiota. Chen et al. [68], using the extrusion-spheronization method, developed 5-ASA microparticles with microcrystalline cellulose and coated them, using an aqueous suspension coating process, with a resistant starch film (23.4% RS2 and 76.6% RS3), obtaining the coated microparticles. RS2 was chosen from a high-amylose cornstarch with 88.5% digestion resistibility. RS3 was prepared by a high-temperature/pressure (HTP) treatment, enzymatic debranching, and retrogradation. In vitro release assays of the mixture RS@MPs showed 40.7% of 5-aminosalicylic acid release within 8 h. In vivo studies performed by oral administration of fluorescein-loaded RS@MPs, indicated their high acidic and enzymatic resistibility and a restrained release in the upper GIT, demonstrated the colon-specificity of the formulation [68].

Some polysaccharides such as pectin (heterosaccharide, generally linear, derived from the cell wall of plants), chitosan (linear polysaccharide composed of randomly distributed β-(1 → 4)-linked D-glucosamine (deacetylated unit), and *N*-acetyl-d-glucosamine (acetylated unit)), guar gum (galactomannan polysaccharide extracted from guar beans), or inulin (a heterogeneous collection of fructose polymers) was extensively used for the development of colonic targeted formulations, because they can be gradually degraded through the colon enzymes, and nowadays there are new and better proposals with these materials. For example, recently, Günter et al. [69] obtained prednisolone-loaded calcium pectinate gel beads, using low methyl-esterified pectins from cell walls of different origins like *S. vulgaris* (SV, SV > 300, *T. vulgare* (TV, TV > 300), *L. minor* (LM, LM > 300), or apple (AP, Classic AU 701). Drug release was investigated in simulated GIT media and the results suggested that the prednisolone release was mainly in the simulated colonic medium, because of the enzymatic erosion of the beads. Based on these results, researchers could conclude that the calcium pectinate gel beads developed are an economic and effective approach for obtaining colon-targeted drug delivery [69].

Disulfide bonds can be selectively broken in the colon. For this reason, molecules containing this bond, such as dextran or ß-galactose, could be selected to obtain specific colon drug delivery to treat IBD or other colon diseases [70]. In this context, Qiao et al. [71] derived a new polymer, containing a hydrophilic molecule (poly(ethylene glycol) (PEG) linked to a hydrophobic one (curcumin (Cur)), through disulfide bonds. In vivo assays carried out with a dextran sulfate sodium(DSS)-induced murine model revealed that polymer administration provided the best reduction of the colonic inflammatory process, compared to sulfasalazine administration [71].

Coating films of amylose associated with other materials such as ethylcellulose or acrylate-based polymers, were found to reach the intact colon and were degraded by colonic bacteria, allowing drug delivery [72]. In this way, a new formulation of prednisolone pellets coated with glassy amylose, in combination with ethyl cellulose (COLAL-PRED^®^), provided excellent results in colonic delivery, and it overcame Phase III trial for clinical use in United Kingdom [73].

Nowadays, new polysaccharides such as arabinoxylans or agave fructans or modified polisaccharides fermentable through colonic microbiota are explored to obtain more specific and stable, targeted colon formulations [74,75,76,77].

The use of prodrugs of smart particles containing an azo bond is a recurrent approach to target colon that, nowadays, is the common strategy in marketed IBD drugs, such as sulfasalazine (as explained above), olsalazine, or balsalazide. This strategy was recently used in the development of more release systems, such as hydrogels. Obtaining hydrogels with polymers or linkers containing azo groups, allows the specific release of drugs in colon, thanks to the breakdown of the azo bonds by the azorreductases produced by the anaerobic flora of the colon. Naeem et al. developed a mixed enzyme/pH-sensitive nanoparticle, using both polymers, enzyme-sensitive and azo-polyurethane, in the synthesis process [78]. Teruel et al. designed and synthetized mesoporous particles loaded with a dye and capped with an azoderivative attached to the external surface of the inorganic matrix through means of a carbamate group. In vitro simulated digestive process assay indicated that the presence of the bulky azoderivative onto the external surface of the mesoporous microparticles inhibited cargo release, whereas in the presence of a reducing agent added to a colonic medium (mimicking reducing environment produced by anaerobic bacterium), cargo delivery was observed [79]. The same group obtained silica mesoporous microparticles loaded with hydrocortisone and capped with an olsalazine azoderivative. In vivo assays revealed that the action of azo-reductases enzymes allows the specific delivery of both 5-ASA (from the molecular gate) and hydrocortisone (located inside of pores), from the microparticles to the colon tissue. As consequence of treatment, injured rats improved the values of disease markers, such as the colon/body weight ratio and the clinical activity score [80].

In the colon, there are more than 10 strains of colonic bacteria, that can produce azorrecturases enzymes such as *Eubacterium hadrum* (2 strains), *Eubacterium* spp. (2 species), *Clostridium clostridiiforme*, a *Butyrivibrio* sp., a *Bacteroides* sp., *Clostridium paraputrificum*, *Clostridium nexile*, and a *Clostridium* sp. [81], as the break of the azo links is guaranteed. Moreover, it was determined that the presence of this bacterial species was not affected in patients with IBD or other colon diseases [15,35,36]. For this reason, this strategy is suitable for IBD patients and allows the selective release in the affected bowel section in pathological conditions. Table 2 summarizes examples following this approach.

However, the enzyme-sensitive approach has some limitations. First, sometimes drug delivery is produced before the formulation reaches the colon due to release that is not specific to the presence of similar enzymes in other digestive sections, Second, in disease conditions, increasing intestinal motility that produces a dramatic reduction of the residence time of formulations in the colon is commonly observed, which can result in an incomplete drug release (this is also the main limitation in pH-sensitive approaches too). Third, sometimes the degradation process of polysaccharides carried out by the colonic enzymes is very slow [88]. Moreover, more in vivo evaluations and studies need to be carried out to validate the specificity of this system in animal models of IBD diseases.

### 5.3. Oral CDDS Following Inflammation-Targeting and ROS-Responsive Approach

As already mentioned above, nanoparticles, due to their size and characteristics, tend to accumulate in inflamed tissues. ROS produced by inflammatory activity, overflow the capacity of the physiological mechanisms of antioxidant protection, and induce oxidative stress, which can cause direct damage to structural cells, amplify inflammation, and induce the formation of pro-inflammatory mediator molecules. High levels of ROS, therefore, makes normal cell functions and cellular signaling difficult and implies an increase in the inflammation process, with consequent aggravation of the symptoms of the disease, such as an increase in mucosal injury, and acceleration of mucosal ulceration in the pathogenesis of IBD [89,90,91,92]. Experimental studies revealed that ROS levels are 10 to 100 times higher in IBD patients than in healthy subjects [92,93,94]. The main ROS involved are superoxide, peroxide, and hydroxyl radical. Vong et al. [95,96] studied the specific accumulation in inflamed colon tissues of redox nanotherapeutics, administered by the oral route and the possibility of taking into account this condition, for IBD and other colon pathology treatments. Following this theory, antioxidant treatments with ROS scavengers were tested to slow the progression of inflammation, by breaking the free radical cascade, in order to improve IBD symptoms [92,97,98,99]. Smart ROS-sensitive formulations could be designed to obtain selective release in inflamed tissues. Following this approach, Zhang et al. obtained a new formulation with a biocompatible ß-cyclodextrin-derivative, loaded with tempol (Tpl/OxbCD). This system is a powerful tool against free radicals as the container is a catalase mimic agent and the drug is a superoxide dismutase mimicker [100]. OxbCD catalyzes the reaction of hydrogen peroxide decomposition in water and oxygen, such that, in these conditions of oxidative stress, tempol is released and neutralizes superoxide radical. This dual catalase/SOD mimetic nanosystem allows us to attack free radicals from different perspectives, allowing great effectiveness in selectively scavenging ROS in inflamed tissues. In vivo studies with this nanosystem were conducted using mice with induced ulcerative colitis. Treatment with Tpl/OxbCD nanoparticles showed that the system is effective in reducing inflammation markers, in improving symptoms and, moreover, showed a safety profile [92]. Another approach to targeting inflamed tissues in IBD, in order to treat ulcerative colitis, was described by Xiao et al. The authors synthetized a CD98 siRNA/curcumin polymeric poly(lactic-co-glycolic) acid/poly(vinyl acid)/chitosan (PLGA/PVA/chit) combination of nanoparticles functionalized with hyaluronic acid (HA-siCD98/CUR-NPs). Previously, the author demonstrated that the cargo, CD98 siRNA (siCD98), produced the down-regulation of colonic CD98 expression and the incorporation of this molecule to the treatment, alleviated the symptoms of UC in a mice disease model [101]. The other drug, curcumin, is a well-known, potent, anti-inflammatory agent. Surface functionalization with hyaluronic acid (HA), helps nanoparticles to specifically bind to glycoprotein CD44, which is over-expressed on the surface of colonic epithelial cells and the macrophages of UC tissues [102,103]. Co-administration of both curcumin and siCD98-based therapies loaded in the HA-functionalized nanodispositives, provided better results than each monotherapy, due to the synergy and summation of effects of the different nanodispositive components. Experimental in vitro and in vivo assays showed a potent reduction of inflammation levels and an important reduction of the disease symptoms [104].

Zhang et al. [105] developed PLGA/Polylactic acid-Polyethylene glycol-Folate nanoparticles (PLGA/PLA-PEG-FA) loaded with 6-shogaol. In vivo assays in which formulation was orally administrated to ulcerative colitis induced mice, showed that formulation targets inflamed tissue and produced a reduction of colitis symptoms and an increased wound repair, by regulating the expression levels of pro-inflammatory and anti-inflammatory factors.

Dou et al. [106] designed a polymeric self-nanoemulsion of Bruceine D, in order to improve the pharmacokinetics parameters and drug effectivity, as compared to the drug suspension. In vivo assays carried out in a trinitrobenzenesulfonic acid (TNBS) colitis-induced rat model, revealed an important alleviation of colitis symptoms, reduction of disease activity index (DAI), and oxidative stress levels and indicators.

Table 3 summarizes these examples and includes more recent findings.

### 5.4. Oral CDDS Following a Dual or Combined Approach

Although some of the simple proposals are clinically successful, especially the pH-dependent formulations, occasionally intact formulations in feces of patients were observed. This might be due because the pH in IBD patients colon is acidified (and as a consequence, the formulations did not release its contents, because the pH required for delivery was not reached) or because the colonic transit time was decreased as a result of diarrhea and the time of exposure of the formulation to the colonic fluids was not sufficient for drug delivery [113,114]. Quick coating dissolution and long residence time are required for a successful and complete colonic drug release, but these parameters are affected by several factors, such as diet, quantity, and characteristics of gastrointestinal fluids, or the disease conditions described above

Limitations of the strategies described stem, mainly, from inter–intraindividual variation and differences between health conditions and disease conditions. In addition, as already described, the disease can manifest itself with varying degrees of severity, making it difficult to ensure the proper functioning of smart strategies. In order to overcome simple strategy limitations, some authors designed treatment strategies combining different approaches. The most studied combinations were (i) pH-/time-sensitive dual systems that depend on pH changes along the GIT, and transit time to release the cargo, and (ii) pH-/enzyme-sensitive dual systems that take advantage of pH changes and different microbiota along the GIT, in order to ensure a targeted drug delivery. In the last decade, very interesting systems were obtained. These were described in an interesting recent review [49].

pH-/time-sensitive dual systems are based on a double coating, in order to protect formulation from drug release, after a certain period of time or until it reaches pH ≈ 6.8–7.4, in order to provide sustained release in the colon. Combinations of different Eudragit™ polymers (described above) allow to put into practice, this double approach. Following this approach, Naeem et al. developed pH-/time-sensitive nanoparticles, using a double layer coating—Eudragit^®^ FS30D coating was include as a pH-dependent polymer, and Eudragit^®^ RS100 as a time-dependent controlled release one. In vitro release studies indicate that most of the drug was released at colonic conditions. In vivo assays revealed that the dual pH/time-dependent nanoparticles treatment allows us to reach specific drug delivery to the inflamed colonic region, increasing the efficacy obtained with the free drugs [115].

In order to overcome the limitations of pH and enzyme-sensitive approaches by themselves, and avoid premature drug delivery or no delivery, polysaccharides and polymers could be combined. In pH-/enzyme-sensitive dual systems, the pH-sensitive polymeric layer, protects the enzyme-responsive coating from the pH conditions found in first segments of GIT. At higher pH values, the pH sensitive layer disintegrates and leaves the accessible enzyme-responsive drug release systems. As an example to illustrate this strategy, Naeem et al. [115] combined an enzyme-sensitive azo-polyurethane (Azo.pu) and a pH-sensitive Eudragit S100 (ES) polymers (ES-Azo.pu to obtain enzyme/pH dual sensitive nanoparticles for specific colon drug delivery [78], and loaded them with coumarin-6 (C-6) (used as a model drug). In vitro release studies revealed that these pH-/enzyme-sensitive nanoparticles avoid drug release in the first segments of the digestive tract characterized by low pH values, and allow an important release in the colonic medium, which included azoreductase enzyme from rat fecal contents (with UC disease). In vivo localization assays indicate that pH-/enzyme-sensitive particles levels were 5.5-fold higher in the inflamed colon than the same particles, but only with a pH sensitive layer coating (pH-sensitive particles) indicating a greater selectivity in the distribution of the dual strategy.

Chitosan is a polycationic biopolymer, which combined with other materials like Eudragits, cellulose, or polyvinyl acetates, provide a potent coating, which has been studied for more than a decade. Recently, Kim et al. [116] developed a new colonic delivery system, consisting of a core tablet including citric acid for acidification, a layer-coating of chitosan-based polymer, and an outer enteric coating combining Eudragit E100^®^ and ethylcellulose.

Ibekwe et al. [117] designed a dual system that combined pH and an enzymatic trigger, commercialized as PhloralTM^®^. This coating combined a polysaccharide in the same coating, to provide the enzymatic trigger and an enteric polymer Eudragit^®^ S and to provide pH trigger when pH = 7 was reached. The efficacy of this combined strategy was demonstrated in healthy human individuals and the system acted as an accurate and fail-safe colonic release disposition, in case one of the approaches was not activated [57,118]. Studies with Phloral were continued by Varum et al. [119], which explored the relationship between the manufacturing process and the behavior of the coating tablets. Moreover, a new 5-ASA formulation based on Phloral technology embedded into OPTICORE^®^ technology was recently developed by this research team [48]. This coating technology consists of combining an alkaline inner layer to accelerate drug release with an enteric outer layer consisting of a mixture of Eudragit S and resistant starch, in order to combine pH and bacterial drug release strategies. OPTICORE was revealed to be an innovative and specific colonic targeting strategy. Both, PhloralTM^®^ and Eudragit^®^ coatings demonstrated to be a potent tool to achieve colon-specific drug delivery.

Zein is a potential option to encapsulate and deliver low solubility drugs to the colon. Recently, a film coating based on a combination of biopolymer Zein with Kollicoat^®^ MAE 100P demonstrated their ability of specific release in the colon. Author studies indicate that the ratio of both coating components and the thickness of the layer are determinant parameters to reach colonic delivery [120].

Aina et al. [121] reported another example of pH-/enzyme-sensitive dual system. In this case, the CDDS relies on the stable release at acid pH medium of 5-ASA from a prodrug consisting of chondroitin sulfate linked to 5-ASA (CS-5-ASA). This product could be used for colon diseases because the conjugate would pass through the stomach and small intestine without alterations, due to the formation of the ester linkage between the drug and the polymer. Additionally, the authors found that the delivery of the drug requires the presence of enzymes like esterases, which are present in the intestinal environment, to complete the reaction that triggers the 5-ASA release. Biodistribution studies showed the mucoadhesion of the product with the intestinal membrane, suggesting this product as a relevant option in colon disease treatment [121].

Hou et al. [122] designed and obtained nanocomposite hydrogels based on azo cross-linked graphene oxide that is suitable for providing specific colon drug delivery after oral administration. This approach could be used to administrate different drugs, thanks to the nanocomposite hydrogels versatility and biocompatibility.

Other dual approaches include the combination of magnetically-driven enzyme-sensitive microparticles. In this study, authors [123] developed hydrocortisone mesoporous microparticles with a magnetic core and an azoderivative molecule as a molecular gate. In vivo assays carried out with rats showed that the efficacy of the site targeting achieved with the azoreductase activity was improved when the rats wore a magnetic belt. The magnetic belt increased the retention time of formulation in the colon and allowed a full drug release. Table 4 summarizes other examples, following the combined approach.

The above examples demonstrate that combined approaches allow us to overcome large limitations of the conventional strategies. However, due to the complexity of IBD, it is necessary to design and develop new strategies for effective and specific colonic drug delivery systems

## 6. Conclusions

Important efforts were made by the scientific community around the world to improve oral CDDS for IBD therapy. Most available remedies for IBD are focused on the remission of symptoms, and the selected drugs to treat IBD depend on the phenotype and severity of the disease. The development of new colonic drug delivery systems (CDDS) has gained interest in the last decades to treat IBD, in order to increase the efficacy of the treatments using smaller amount of active agents, reduce adverse effects of the drugs by reducing intestinal drug absorption, and reduce the frequency of drug administration. In fact, some reported systems using polymer coatings already approved by drug agencies and administrations could be eligible for a quick entrance into the market. Reports are mainly focused on the development of CDDS in which drug release is triggered by pH, the presence of specific enzymes or ROS. However, this single approach exhibits limitations due to the inter- and intrapersonal variability and due to the pH, the fed condition, transit time, and microbiota variations in patients. For these reasons, researchers work in new formulations combining materials and strategies to achieve complete and specific colonic delivery. Some examples of dual strategy systems, such as PhloralTM^®^ or OPTICORE^®^ technology allow us to obtain promising results, even in specific delivery and effectively, even in disease conditions. Despite many CDDS being successfully developed to reach their target, further research work it is still necessary to elucidate the molecular basis of inflammatory bowel diseases, in order to design specific therapies and site-targeted systems that allow full health recovery by restoring the immunity system, as well as the gut microbiota, in order to improve the quality of the treatments and life of IBD patients. Moreover, a new challenge would be to obtain functionalized nanoparticle formulations that specifically internalize into the cells of inflamed tissue and specifically release inside the target cells, in order to increase the drug levels in damaged tissues, at reduced doses. These systems could be combined with probiotic administration, in order to accelerate patient recovery and prevent relapses. The development of new drugs and new formulation strategies for treatment is, in turn, subjected to the acquisition of greater knowledge about the etiology and pathophysiology of these diseases. New knowledge will allow us to design and obtain more effective treatments. Much more research is needed in all fields around IBD (physiology, pharmacology, pharmaceutical technology, etc.) and, above all, to enhance the transfer of most promising preclinical results to clinical trials and, subsequently to the market, to be able to provide a complete therapy to these patients, which allows them to be cured or, at least, improve their quality of life much more.

## Figures and Tables

**Figure 1 ijms-21-06502-f001:**
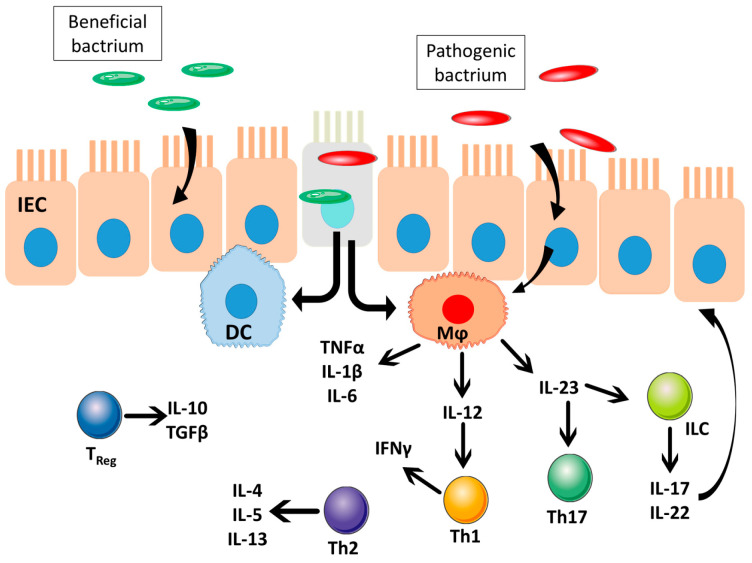
Key cell populations and mediators of bowel homeostasis and pathogenesis of inflammatory bowel disease. DC—dendritic cells, IEC—intestinal epithelium cells, ILC—innate lymphoid cells, Mφ—macrophages, sIgA—secreted IgA, Th—T helper cells, and Treg—T-cell regulating. Adapted from Mathisen [9].

**Figure 2 ijms-21-06502-f002:**
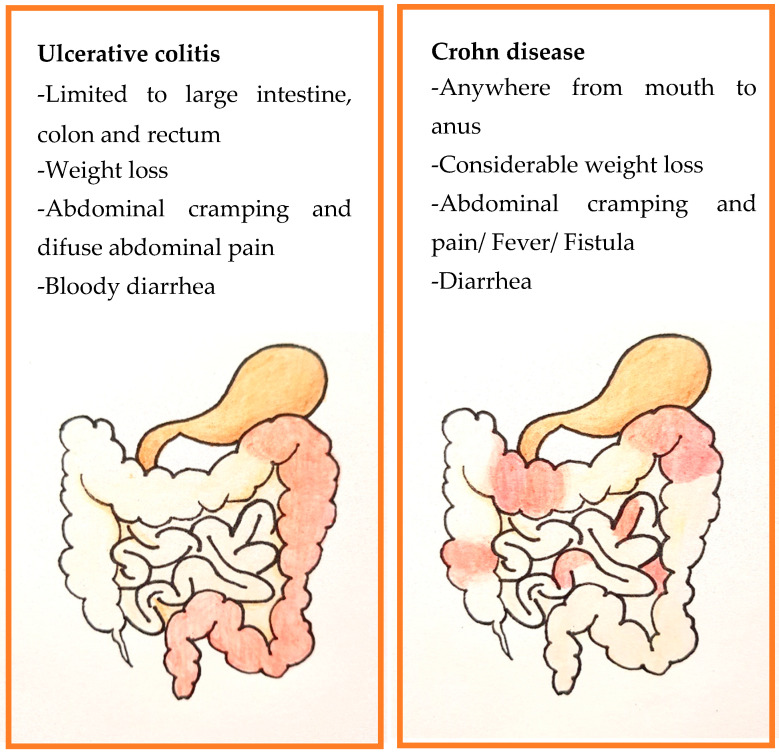
Gastrointestinal tract (GIT) affected by the Inflammatory Bowel Disease. Left—GIT showing ulcerative colitis. Right—GIT showing Crohn’s disease.

**Figure 3 ijms-21-06502-f003:**
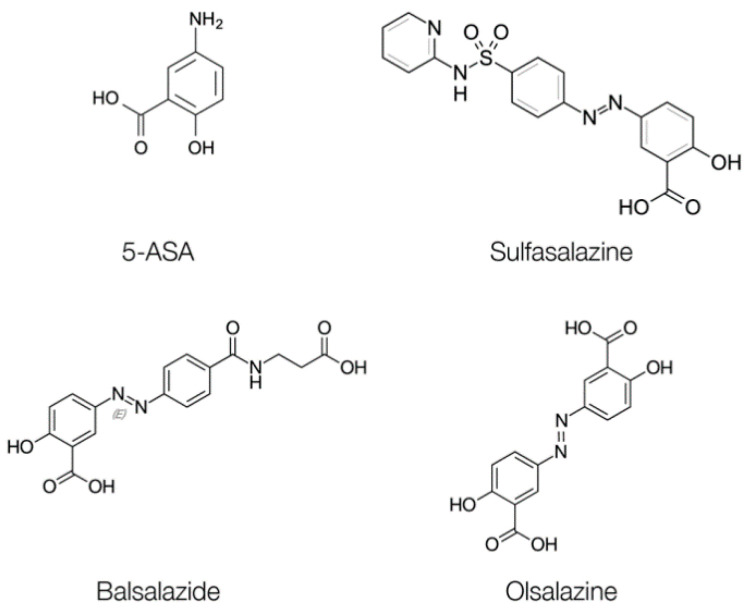
Chemical structures of the aminosalicylate drugs 5-ASA, sulfasalazine, balsalazide, and olsalazine.

**Table 1 ijms-21-06502-t001:** Oral colonic drug delivery systems (CDDS) following pH-sensitive approach.

Approach/Mechanism	Dosage Form	Description	Drug Loaded	In Vitro Model and Release	In Vivo Model	Year/Ref
pH-responsive	Nanofibers: Polyvinylpyrrolidone (PVP).	Diclofenac encapsulated in a hydrophilic nanocomposite coated with a thin shellac layer. The nanosystem was obtained using a modified coaxial electrospinning method and showed specificity and pulsatile colon delivery.	Diclofenac sodium	Drug release studies in buffered solutions at different pH and ex vivo permeation studies.	-	2018 [15]
pH-responsive	NPs: Eudragit FS30D/PLGA nanoparticles	Nanoparticles of cyclosporine A encapsulated with Eudragit FS30D nanoparticles (ENPs), PLGA nanoparticles (PNPs), and Eudragit FS30D/PLGA nanoparticles (E/PNPs) were synthetized. In vivo assays using a mouse model of colitis, showed a significant improvement of the symptoms of the pathology	Cyclosporine A	Drug release studies in buffered solutions at different pH.	Mice Dextran Sulfate Sodium (DSS) induced colitis	2018 [42]
pH-responsive	Microcrystals: Chitosan/alginate/Eudragit S multilayers	Dexamethasone microcrystals coated with different layers: chitosan, alginate, and Eudragit S 100 (ES) allows a pH-dependent dexamethasone release, providing an important drug release in colon segment. The formulation shows therapeutic activity in an in vivo mouse model of ulcerative colitis.	Dexamethasone	Drug release studies in buffered solutions at different pH.	Mice DSS induced colitis	2018 [43]
pH-responsive	NPs: Chitosan based polymeric nanomicelles	pH-sensitive nanomicelles of curcumin coated with N-naphthyl-N,O-succinyl chitosan (NSCS) and N-octyl-N,O- succinyl chitosan (OSCS) allow colon-targeted drug delivery.	Curcumin	Drug release studies with simulated GIT fluids. Cytotoxicity: Caco-2 and HT-29 cells. Anti-cancer activity: HT-29 cells.	-	2018 [23]
pH-responsive	Nanostructured lipid carrier (NLCs) systems coated with Eudragit S100	Budesonide loaded nanostructured lipid carriers obtained by high pressure homogenization technique and coated with Eudragit^®^ S100 prevent release of drug at acidic pHs.	Budesonide (BDS)	Drug release studies in buffered solutions at different pH.	-	2018 [27]
pH-responsive	NPs: Mesoporous nanoparticles capped with (1) a molecule containing a disulfide bond, (2) a starch derivative or (3) a lipid bilayer.	Gated mesoporous nanoparticles able to deliver their cargo triggered by different stimuli such as redox ambient, enzymatic hydrolysis, and presence of surfactant or contact with cell membrane) allow an increase of drug release and drug concentration in intestine and colon reducing plasma drug levels.	Safranin O (dye for preliminary studies)	Drug release studies with simulated GIT fluids. Cytotoxicity & permeability: Caco-2 cell line.	Wistar rats Biodistribution studies (S2 coated with Eudragit FS30D)	2017 [64]
pH-responsive	MPs: Alginate hydrogel/chitosan micelle composites	pH-sensitive drug delivery systems based on the cross-linked unimolecular micelles dispersed in a hydrogel matrix at different ratios. The hydrogel/micelle (3:1) showed a colon-specific release. The release of drug from these formulations is a complex process that involves several mechanisms simultaneously.	Emodin	Drug release studies with simulated GIT fluids.	-	2018 [61]
pH-responsive	Alginate microparticles	pH-sensitive drug delivery microparticles of puerarin in order to reduce the complications associated to ulcerative colitis	Puerarin	Drug release studies with simulated GIT fluids	Mice DSS-induced colitis-associated colorectal cancer	2019 [65]
pH-responsive	MPs: P(LE-IA- MEG) hydrogel microspheres obtained using emulsion crosslinking method with itaconic acid, poly (ethylene glycol) methyl ether methacrylate and PLE-AC (a new polymer obtained from poly (ethylene glycol) methyl ether methacrylate and acryloyl choride)	pH-sensitive hydrogel microspheres as a drug carrier of hydrocortisone have been obtained. The in vitro assays showed that the microspheres are pH-sensitivity. In vivo assays carried out on a mice colitis model indicate that mice treated with microspheres exhibited more therapeutic effects than those treated with free hydrocortisone.	Hydrocortisone sodium succinate (HSS)	Drug release studies in buffered solutions at different pH.	Mice 2,4,6-Trinitrobenzenesulfonic acid (TNBS) induced colitis	2018 [66]
pH-responsive	Tablet of aloe vera polysaccharide/Acrylonitrile nanoparticles + guar gum + drug	Self-assembled nanoparticles with aloe vera polysaccharide and acrylonitrile loaded with 5-ASA were synthetized. The resulting allows the cargo delivery at colonic pH.	5-ASA	Drug release studies in buffered solutions at different pH.	-	2018 [67]
pH-responsive	Eudragit microparticles	Celecoxib microparticles formulated using Eudragit S100, Eudragit L100-55 or Eudragit L100	Celecoxib	In vitro and in vivo assays	Rat model	2016 [52]
pH-responsive	ColoPulse coated tablets	Sustained-release budesonide tablets coated with ColoPulse	Budesonide	In vitro and in vivo assays	Human	2019 [62]
pH-responsive	ColoPulse coated tablets	Sustained-release infliximab tablets coated with ColoPulse	Infliximab	In vitro assays		2019 [63]

**Table 2 ijms-21-06502-t002:** Oral CDDS following the enzyme-sensitive approach.

Approach/Mechanism	Dosage Form	Description	Drug Loaded	In Vitro Model & Release	In Vivo Model	Year/Ref
Enzyme-responsive	Microparticles: Starch film-coated	Microparticles loaded with 5-ASA coated with a resistant starch films prepared with different techniques showed an important enzymatic resistance. In vivo studies indicated that formulation orally administered resist acidic medium and delivered drug in colon	5-ASA	-	Healthy mice	2018 [68]
Enzyme-responsive	Microspheres: guar gum and xanthan gum	5-ASA encapsulated in microspheres with guar gum and xanthan gum was combined with probiotics such as *Lactobacillus acidophilus*, *L. rhamnosus*, *Bifidobacterium longum*, and *Saccharomyces boulardi*. The coadministration of probiotics and drugs show therapeutic benefits in the in vivo assays carried out in rats with ulcerative colitis.	5-ASA	-	Wistar rats. Acetic acid- induced ulcerogenic colitis.	2017 [82]
Enzyme-responsive	MPs: Mesoporous silica microparticles capped with a bulky azo derivative	Gated silica microparticles loaded with the dye safranin O were prepared and characterized. Microparticles release takes place under reducing conditions typical of the colonic mucosa. Preliminary in vivo experiments using healthy mice indicate that solid release the dye in the last part of GIT mucosa.	Safranin O (dye for preliminary studies)	Drug release studies in buffered solutions at different pH (with/without enzyme stimuli).	Healthy mice	2018 [83]
Enzyme-responsive	MPs: Magnetic mesoporous silica microparticles capped with a bulky azo derivative	Magnetic micro-sized mesoporous silica particles loaded with safranin O and functionalized with an azo derivative allow colon-targeted delivery. Controlled release assays were carried out using simulated digestion process.	Safranin O (dye for preliminary studies)	Drug release studies with simulated GIT fluids.	-	2018 [79]
Enzyme-responsive	NPs: Star-shape amphiphilic polymer of polycaprolactone (PCL), olsalazine, and methoxypolyethylene glycols (mPEG).	Azo four-arm polymeric micelles for colon-targeted delivery of dimethyl fumarate can be use in colon cancer therapy. In vitro drug release assays indicated that the cumulative drug release from the polymeric micelles was lower than 20% in the gastric fluid of rats within 10 h. However release in colonic fluids of drugs reached 100% in the same period of time	Dimethyl fumarate	LIVE/DEAD^®^ Viability/Cytotoxicity Assay Kit—Colon cancer cell lines CT26, HT29, and HCT116 cells.	-	2016 [84]
Enzyme-responsive	MPs: Calcium pectinate gel beads	Low methyl-esterified pectins microparticles were used as the carriers for colon delivery of prednisolone. Release aspects of prednisolone in the simulated gastric (pH 1.25), intestinal (pH 7.0) and colonic (pH 7.0 + pectinase) media were investigated. Prednisolone release occurred in a larger extent in colonic medium due to the enzymatic erosion of the beads.	Prednisolone	Drug release studies with simulated GIT fluids.	-	2016 [69]
Enzyme-responsive	Pellets (0.7–1 mm) +microbiota sensitive film coating	5-ASA pellets coated with Nutriose: ethylcellulose 1:4 or peas starch:ethylcellulose 1:2 blends were synthetized. In vivo assays in a rat model revealed the efficacy of these colon targeting pellets.	5-ASA	-	Wistar rats TNBS induced colitis	2015 [85]
Enzyme-responsive	NPs: MSNs guar gum capping (GG-MSN)	In vitro release studies of 5-Fluorouracil loaded mesoporous nanoparticles capped with guar gum allow drug release specifically triggered by colonic enzymes. The released drug produced cytostatic effect in cells cultured with simulated colonic microenvironment	5FU	Drug release studies with simulated GIT fluids Cell proliferation study human colon cancer model (HT-29).	-	2017 [86]
Enzyme-responsive	NPs: Amphiphilic curcumin polymer (PCur)	An amphiphilic curcumin polymer conjugate containing a hydrophilic poly(ethylene glycol) and hydrophobic curcumin linked by disulfide bond was designed to release curcumin in the intestinal reduction environment. The results obtained from in vitro and in vivo assays confirmed higher therapeutic level of curcumin in intestinal tissue damaged.	Curcumin	Drug release studies in buffered solutions at different pH and reductive environment. Cytotoxicity and permeability: Caco-2 cells.	Sprague–Dawley (SD) rats and C57BL/6 mice DSS-induced IBD model	2017 [71]
Enzyme-responsive	Tablets: Chitosan-laurate coating	Chitosan-laurate dispersions were used as coating films of acetaminophen tablets. Results indicate that formulation was stable in acidic environment and allow drug delivery in the colon tissues.	Acetaminophen	Drug release studies in buffered solutions at different pH (with/without enzyme stimuli).	-	2015 [87]

**Table 3 ijms-21-06502-t003:** Oral CDDS following inflammation targeting and ROS-responsive approach.

Approach/Mechanism	Dosage Form	Description	Drug Loaded	In Vitro Model & Release	In Vivo Model	Year/Ref
Inflammation targeting (ROS-responsive)	NPs: An oxidation-responsive β-cyclodextrine nanoparticles loaded with Templol (Tpl/OxbCD)	An oxidation-responsive β-cyclodextrine (OxbCD) nanoparticles was obtained and loaded with tempol (Tpl). The drug release from formulation is allowed by hydrolysis of OxbCD NPs by means of hydrogen peroxide. Oral administration of nanoparticles allows to obtain increased amounts of drug in colon and less biodistribution in other organs. The efficacy was better than free drug or other tested formulations used as control	Tempol (Tpl): ROS scavenger anti-inflammation	ROS-sensitivity, hydrolysis and drug release evaluation in buffered solutions at different pH.	Mice DSS and TNBS induced colitis	2016 [92]
Inflammation targeting	NPs: PLGA	PLGA NPs loaded with siCD98 and curcumin demonstrated that codelivery of both drugs increase the efficacy of colitis treatment. This structurally simple platform is suitable for orally administered delivery of drugs to target colon for ulcerative colitis or other pathologies	CD98 siRNA and curcumin	-	Mice DSS induced colitis	2016 [101]
Inflammation targeting	NPs: Ultra-small solid archeolipid nanoparticles	Archeolipid nanoparticles loaded with dexamethasone are synthetized and characterized	Dexamethasone (Dex)	Anti-inflammatory activity: J774A1 cells.	-	2017 [107]
Inflammation targeting	NPs: PLGA and amphiphilic copolymer Polylactic acid-Polyethylene glycol-Folate (PLGA/PLA- PEG-FA) nanoparticles	PLGA/PLA-PEG-FA nanoparticles containing 6-shogaol are a promising formulation because its effectivity in targeting colitis tissue, improving symptoms and accelerating wound repair.	6-shogaol	-	Mice DSS induced colitis	2018 [105]
Inflammation targeting	NPs: ginseng-derived nanoparticles (GDNPs 2)	GDNPs 2 nanoparticles ability for controlled and it is an optimal option for ulcerative colitis prevention and treatment due to its effectivity in colon-targeted, low toxicity and easy production.	Lipids, proteins, microRNAs and ginger bioactives (6-gingerol and 6-shogaol)	Internalization and citotoxicity: RAW 264.7 microphage, Caco-2BBE and Colon-26 cells.	Mice DSS induced colitis	2016 [108]
Inflammation targeting	NPs: PLGA nanocarriers	PLGA nanocarriers loaded with cyclosporine A have demonstrated to be useful as drug delivery system, targeting inflamed issues, providing high drug concentrations at inflamed tissues, demonstrating superior efficacy and safety in a relevant preclinical mouse model in vivo.	Cyclosporine A	Drug release studies with simulated gastric fluid (pH = 3.0).	Balb/C mice DSS-induced acute colitis	2017 [109]
Inflammation targeting	NPs: Cationic lipid-assisted nanoparticles (CLAN)	Cationic lipid-assisted nanoparticles loaded with Tacrolimus (FK506) are tested as drug delivery system for ulcerative colitis treatment Results indicate that the formulation accumulate at the inflamed issues and improve the therapeutic effects of the treatment.	Tacrolimus (FK506)	Drug release studies in buffered solutions at different pH.	C57BL/6 mice DSS-induced acute colitis	2018 [110]
Inflammation targeting (ROS-responsive)	NPs: Self-assembling copolymer nanoparticles	Redox nanoparticles (RNPO) administered by oral route specifically accumulated in inflamed tissues and scavenged reactive oxygen species (ROS) demonstrating the potential therapeutic of this approach.	Tempol (Tpl): ROS scavenger anti-inflammation	Cellular uptake: Caco-2 cells.	Mice DSS-induced acute colitis	2015 [95]
Inflammation targeting (ROS-responsive)	NPs: Superoxide dismutase (SOD)/catalase mimetic nanosystem	An oxidation-responsive ß-cyclodextrin material (OxbCD) was synthesized, and loaded with the ROS scavenger Tempol. Hydrogen peroxide presence promote the on-demand release of loaded drug. In vivo assays revealed that nanoparticles accumulate in the inflamed tissues after oral delivery.	Tempol (Tpl): ROS scavenger anti-inflammation	Drug release studies in buffered solutions at different pH + hydrogen peroxide.	DSS induced acute and chronic colitis. & TNBS induced acute colitis	2016 [92]
Inflammation targeting (redox-responsive)	NPs based on 4-aminothiophenol-carboxymethyl inulin conjugate	Budesonide loaded nanoparticles based on an amphiphilic inulin derivative (ATP-CMI) were obtained. In vitro release and in vivo assays indicate that this formulation can be a promising option for colitis treatment.	Budesonide	Drug release studies with simulated GIT fluids. Cytotoxicity: Caco-2 cell line.	Mice DSS-induced acute colitis	2018 [111]
Inflammation targeting (NPs intrinsic properties)	Nanoemulsion: Self-nanoemulsifying drug delivery system (SNEDDS)	Self-nanoemulsifying formulation that contains medium-chain triglycerides oil (MCT oil), Solutol HS-15 (surfactant), propylene glycol (co-surfactant) and Bruceine D is able to show high therapeutic effectivity in a colitis animal model.	Bruceine D	Drug release studies in buffered solutions at different pH.	Sprague Dawley rats TNBS-induced colitis	2018 [106]
Inflammation targeting	NPs: Broccoli-Derived Nanoparticles	Nanoparticles based on broccoli extracts have demonstrated to protect mice against colitis Assays have been carried out using three mouse colitis models and preliminary studies indicate that activation of adenosine monophosphate- activated protein kinase (AMPK) in dendritic cells (DCs) play an important role y in prevention.	BDN	-	1. Adoptive T cell transfer chronic colitis. 2. DSS-induced colitis. 3. Agonistic αCD40 colitis.	2017 [112]

**Table 4 ijms-21-06502-t004:** Oral CDDS following dual or combined approaches.

Approach/Mechanism	Dosage Form	Description	Drug Loaded	In Vitro Model & Release	In Vivo Model	Year/Ref
Dual pH-/time-responsive	NPs: Eudragit FS30D, Eudragit RS100	Loaded budesonide nanoparticles using Eudragit^®^ FS30D as a pH-responsive polymer, and Eudragit^®^ RS100 as a time-dependent controlled release polymer were obtained in order to reach release at a colonic pH. In vivo assays confirmed that the dual approach pH/time-dependent is useful to obtain colon specific delivery and to enhance the efficacy of budesonide treatment.	Budesonide	Drug release studies in buffered solutions at different pH.	Mice DSS induced colitis	2015 [115]
Swelling properties & enzyme-responsive	Microspheric vehicle: Cationic konjac glucomannan (cKGM) phytagel	Microspheric particles obtained with cationic konjac glucomannan phytagel are able to target colonic macrophages and suppress the local expression of TNF-α by specific delivery of antisense oligonucleotide anti-TNFα, providing excellent results in the in vivo assays using a colitis mice model.	Antisense oligonucleotide anti-TNFα	Drug release studies in buffered solutions at different pH. Cytotoxicity: Raw 264.7 and CT-26 cell lines.	Mice DSS induced colitis	2015 [121]
Dual pH-/time-responsive	ColoPulse coated tablet (croscarmellose sodium enhance disintegration)	Infliximab incorporated in a sugar glass matrix showed activity compared to a fresh infliximab solution and demonstrated advantages such as high stability and targeted colon delivery	Infliximab	Drug release studies in buffered solutions at different pH.	-	2016 [124]
Dual pH-/enzyme-responsive	Microspheres: polyacrylamide-graft-gum karaya pH-sensitive spray dried microspheres (PAAm-g-GK)	Microspheres based on pH-sensitive PAAm-g-GK copolymer having cross-linked with glutaraldehyde and loaded with capecitabine are used as drug carriers to target colon tissue. In vitro results indicate that, after 5h later to star the assay, it is observed an important drug release because of colonic bacteria’s action on PAAm-g-GK copolymer contained in fecal contents medium accelerated drug delivery.	Capecitabine	Drug release studies with simulated GIT fluids and rat cecal contents.	-	2017 [125]
Dual pH-/enzyme-responsive	Nanocomposite hydrogel based on graphene oxide pH-sensitive and biocompatible graphene oxide (GO) containing azoaromatic crosslinks and poly (vinyl alcohol) (PVA) (GO–N = N–GO/PVA)	Nanocomposite hydrogel based on graphene oxide, azoaromatic crosslinks, and polyvinyl alcohol, and loaded with curcumin designed for colon cancer drug delivery. The results demonstrated that the nanocomposite hydrogels are able to protect curcumin from acidic pHs and enhance drug concentration and residence time in the colon tissue.	Curcumin	Drug release studies in buffered solutions at different pH.	Healthy Sprague-Dawley rats Gastrointestinal distribution, imaging analysis and PK studies	2016 [122]
Dual pH-/enzyme responsive	Phloral^TM^	Unique patented coating that include Eudragit^®^ and a resistant starch	5-ASA	Drug release at different pH and in pH = 6.8 human fecal slurry		2020 [119]
Dual pH-/enzyme responsive	OPTICORE^TM^	Coating technology consisting on an inner layer of Duocoat^®^ to accelerate the release and an outer layer of Phloral (pH and enzyme responsive coating)	5ASA	Drug release at different pH and in pH = 6.8 human fecal slurry		2020 [48]
pH-responsive & target of CD44 receptors (HA-CD44)	NPs: Hyaluronic Acid-Functionalized nanoparticles encapsulated in a hydrogel of alginate and chitosan (7:3).	Tripeptide lysine-proline-valine was loaded into polymeric nanoparticles obtained from functionalized hyaluronic acid. These nanoparticles are nontoxic and biocompatible with intestinal cells an oral administration of the formulation allow the alleviation of colitis symptoms combining both accelerating mucosal healing and reducing inflammation.	Tripeptide KPV (Lysine-proline-valine)	Cytotoxicity and cellular uptake: Raw 264.7 and Colon-26 cells.	Mice DSS induced colitis	2017 [126]
Magnetically-driven & pH-responsive microparticles	Magnetic mesoporous microparticles with azo-derivative molecular gate	Hydrocortisone magnetic mesoporous microparticles decorated with bulky azo-derivatives allows a colon specific delivery and high in vivo efficacy in a rat model	Hydrocortisone	In vitro and in vivo assays	Sprague Dawley rats TNBS-induced colitis	2018 [123]

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
