# Peer review of "New Insights of Oral Colonic Drug Delivery Systems for Inflammatory Bowel Disease Therapy"

_ijms, 2020, doi:10.3390/ijms21186502_

Round 1
Reviewer 1 Report
This paper describes about ‘New insights of Oral Colonic Drug Delivery Systems for inflammatory bowel diseases therapy’. You reviewed regarding the newest colon target research to the inflammatory bowel disease (IBD) for a decade. You summarized it well, although English problem and careless mistake were recognized a little. Furthermore, I point out the issues to be addressed.
Major
- English problem was recognized. For instance, page 7 line 257, Table 1 summarizes …. It should be it was summarized that …… in Table 1. Page 10, line 378-379, In last decade has…..
- On page 2, line 45, I recommend one article regarding permeability in UC patients. You had better cite this article.
- Comparison of the Intestinal Drug Permeation and Accumulation Between Normal Human Intestinal Tissues and Human Intestinal Tissues With Ulcerative Colitis. Nakai D, Miyake M, Hashimoto A, J Pharm Sci. 2020; 109(4):1623-1626.
- In page 2, you mentioned the difference between CD and UC from lots of angles. But, I think you should mention from the aspect of immunology. As you know and already mentioned UC was observed only in colon. However, CD occurs diffusely. Therefore, the human immunity would be at least involved in the appearance of CD.
- In page 5, line 143-145. You mentioned NPs characteristics of enhancing epithelial permeability and retention. Is that correct? I know well about EPR effect to the cancer treatment. Does it fit to IBD, too?
Minor
- In page 4, line 150, dot should be omitted.
- In page 6, line 195, you should correct from formulacions into formulations.
- In page 6, line 224, you should added at dot after et al.
- In page 8, line 300, you should added at dot after reference.
Author Response
This paper describes about ‘New insights of Oral Colonic Drug Delivery Systems for inflammatory bowel diseases therapy’. You reviewed regarding the newest colon target research to the inflammatory bowel disease (IBD) for a decade. You summarized it well, although English problem and careless mistake were recognized a little. Furthermore, I point out the issues to be addressed.
Thank you very much for you exhaustive revision. Your feedback has been a great help in improving the quality of the article.
Major
- English problem was recognized. For instance, page 7 line 257, Table 1 summarizes …. It should be it was summarized that …… in Table 1. Page 10, line 378-379, In last decade has…..
Changes have been done.
- On page 2, line 45, I recommend one article regarding permeability in UC patients. You had better cite this article.
Comparison of the Intestinal Drug Permeation and Accumulation Between Normal Human Intestinal Tissues and Human Intestinal Tissues With Ulcerative Colitis. Nakai D, Miyake M, Hashimoto A, J Pharm Sci. 2020; 109(4):1623-1626.
Thank you for the suggestion. Comment about permeability has been included and the article that you have suggested has been cited
- In page 2, you mentioned the difference between CD and UC from lots of angles. But, I think you should mention from the aspect of immunology. As you know and already mentioned UC was observed only in colon. However, CD occurs diffusely. Therefore, the human immunity would be at least involved in the appearance of CD.
Details of immunity and serological markers are included in the text
- In page 5, line 143-145. You mentioned NPs characteristics of enhancing epithelial permeability and retention. Is that correct? I know well about EPR effect to the cancer treatment. Does it fit to IBD, too?
Yes, it is a similar effect. It has been included in the text
Minor
- In page 4, line 150, dot should be omitted. Done
- In page 6, line 195, you should correct from formulacions into formulations.
Done
- In page 6, line 224, you should added at dot after et al. Done
- In page 8, line 300, you should added at dot after reference. Done
Reviewer 2 Report
This paper reports a review about recent contributions to improve the effectiveness of treatments of inflammatory bowel diseases using different Colonic Drug Delivery Systems (CDDS). The review is comprehensive and encompasses information from the pathogenesis of the disease, different possibilities of drug delivery systems and finally future perspectives about the topic. Therefore, it is recommended for publication in International Journal of Molecular Sciences after major revision indicated below.
GENERAL COMMENTS
Revise the English of the manuscript. Correction of the manuscript by a native speaker with expertise in the field of chemistry is strongly recommended.
I believe that using some images from other papers for a deeper and clearer understanding of the problem of inflammatory bowel diseases therapies would be beneficial for this review.
Ensure that the authors have permission (copyright) for the inclusion of some images from other papers.
If possible include more references of the journal.
SPECIFIC COMMENTS
The Introduction section should be expanded including more general information about the main topic of the review.
The review lacks on describing the systems more appropriately, such as their chemical structures (synthetic polymers, for example).
There are plenty of unexplained abbreviations, present especially in the Tables. Please unify the nomenclature of chemical compounds, explain all abbreviations.
In my opinion, tables constitute a significant part of publications, it is worth moving some of the information directly to the main text. Tables are not enough informative or too informative. Please revise the Description part of the Tables so that the content is proportional.
Include more possibilities for further studies as future perspectives.
Author Response
This paper reports a review about recent contributions to improve the effectiveness of treatments of inflammatory bowel diseases using different Colonic Drug Delivery Systems (CDDS). The review is comprehensive and encompasses information from the pathogenesis of the disease, different possibilities of drug delivery systems and finally future perspectives about the topic. Therefore, it is recommended for publication in International Journal of Molecular Sciences after major revision indicated below.
Thank you for reviewing our paper and many thanks for your comments
GENERAL COMMENTS
Revise the English of the manuscript. Correction of the manuscript by a native speaker with expertise in the field of chemistry is strongly recommended.
Following your suggestion, English of the manuscript has been revised
I believe that using some images from other papers for a deeper and clearer understanding of the problem of inflammatory bowel diseases therapies would be beneficial for this review. Ensure that the authors have permission (copyright) for the inclusion of some images from other papers.
Figures from other reviews has been included and authors have permission for use them
If possible include more references of the journal.
Done.
SPECIFIC COMMENTS
The Introduction section should be expanded including more general information about the main topic of the review.
Done. Introduction section has been expanded with general information of interest
The review lacks on describing the systems more appropriately, such as their chemical structures (synthetic polymers, for example).
Systems has been explained more conveniently
There are plenty of unexplained abbreviations, present especially in the Tables. Please unify the nomenclature of chemical compounds, explain all abbreviations.
Abbreviations has been explained or substituted by the name.
In my opinion, tables constitute a significant part of publications, it is worth moving some of the information directly to the main text. Tables are not enough informative or too informative. Please revise the Description part of the Tables so that the content is proportional.
Done. Part of the information of the tables has been moved to the main text and the information in the text has been expanded for clarity. Tables has been maintained as a summary
Include more possibilities for further studies as future perspectives.
More possibilities and further studies has been included following your suggestion
Round 2
Reviewer 1 Report
I have no more comment.
Reviewer 2 Report
Authors reviewed carefully the manuscript following all the suggestions previously commented. Therefore, I recommend this manuscript for publication in International Journal of Molecular Sciences.